# Assessment of the Best FEA Failure Criteria (Part II): Investigation of the Biomechanical Behavior of Dental Pulp and Apical-Neuro-Vascular Bundle in Intact and Reduced Periodontium

**DOI:** 10.3390/ijerph192315635

**Published:** 2022-11-24

**Authors:** Radu Andrei Moga, Stefan Marius Buru, Cristian Doru Olteanu

**Affiliations:** 1Department of Cariology, Endodontics and Oral Pathology, School of Dental Medicine, Iuliu Hatieganu University of Medicine and Pharmacy, Str. Motilor 33, 400001 Cluj-Napoca, Romania; 2Department of Structural Mechanics, School of Civil Engineering, Technical University of Cluj-Napoca, Str. Memorandumului 28, 400114 Cluj-Napoca, Romania; 3Department of Orthodontics, School of Dental Medicine, University of Medicine and Pharmacy Iuliu Hatieganu Cluj-Napoca, Str. Avram Iancu 31, 400083 Cluj-Napoca, Romania

**Keywords:** periodontal breakdown, FEA failure criterion, maximum hydrostatic pressure, orthodontic movement, dental pulp, neuro-vascular bundle

## Abstract

The aim of this study was to biomechanically assess the behavior of apical neuro-vascular bundles (NVB) and dental pulp employing Tresca, Von Mises, Pressure, S1 and S3 failure criterions in a gradual periodontal breakdown under orthodontic movements. Additionally, it was to assess the accuracy of failure criteria, correlation with the maximum hydrostatic pressure (MHP), and the amount of force safe for reduced periodontium. Based on cone-beam computed tomography, 81 3D models of the second lower premolar were subjected to 0.5 N of intrusion, extrusion, rotation, tipping, and translation. A Finite Elements Analysis (FEA) was performed. In intact and reduced periodontium apical NVB, stress (predominant in all criteria) was significantly higher than dental pulp stress, but lower than MHP. VM and Tresca displayed identical results, with added pulpal stress in translation and rotation. S1, S3 and Pressure showed stress in the apical NVB area. 0.5 N seems safe up to 8 mm periodontal breakdown. A clear difference between failure criteria for dental pulp and apical NVB cannot be proved based only on the correlation quantitative results-MHP. Tresca and VM (adequate for ductile materials) showed equivalent results with the lowest amounts of stress. The employed failure criteria must be selected based on the type of material to be analyzed.

## 1. Introduction

Orthodontic therapy and its effects over the apical neurovascular bundle (NVB), dental pulp and periodontal ligament (PDL) in a reduced periodontium has been insufficiently studied [1,2,3,4,5,6,7,8,9,10]. Even a low orthodontic force, considered to be safe when applied in the intact periodontium, could alter the circulatory pressure balance from these tissues in a reduced periodontium (by exceeding the maximum physiological hydrostatic pressure [MHP] of 2–16 KPa that prevents capillary occlusion) due to different types of localized stresses (i.e., tensile, compressive, shear, overall and hydrostatic pressure) [1,2,3,4,5]. As consequence, ischemia, pulpal necrosis, further periodontal loss, and orthodontic root resorption could shortly follow t orthodontic therapy due to the anatomical and functional interconnectivity of the periodontium’s anatomical components [1,2,3,4,5,6,7]. The optimal amount of orthodontic force (i.e., acknowledged to be light and producing the desired movement without tissue damage) for an intact periodontium remains a subject of debate [1,2,3,4,5,6,7,8,9,10,11]. There are reports for intact periodontium that 0.5–0.6 N to be safely applied for periodontal ligament, while more than 0.6 N may produce stress exciding MHP, thus inducing regressive changes [1,2,3,4,8,9,10,12,13]. A recent in vitro study reported for the cervical third of PDL that 0.5 N is safe to be used in the intact periodontium, while the amount of force should be reduced to 0.1–0.2 N for rotation, 0.15–0.3 N for translation and 0.2–0.4 N for tipping in 4–8 mm periodontal breakdown [1]. Nonetheless, for the apical third of PDL the same study [1] reported 0.5 N to be safely applied on up to 8 mm of bone loss. No studies with focus on the NVB and dental pulp during orthodontic movements have been found, except for our earlier work [4]. Two review studies suggested a poor quality for multiple in vivo research with focus on PDL in the intact periodontium and asked for further investigations of this subject [5,9]. When investigating the reduced periodontium and the optimal amount of orthodontic force to be safely applied, less information is available, while the risks of further ischemic complications and tissue damage is even more important than in the intact periodontium. Thus, it is acknowledged that the MHP must not be exceeded (by the amount of stress produced by the orthodontic movement), and that it seems to play a significant role in the study of both the intact and reduced periodontium [1,2,7].

Orthodontic biomechanical behavior in periodontal breakdown is impossible to be studied other than using in vitro methods (i.e., Finite Elements Analysis [FEA]) due to the complexity of anatomical structures, multiple variables, and the complexity of the process [1,2,3,4]. FEA is a mathematical based method which is widely used in the engineering field and has been adapted in the last decades for medicine and dentistry, allowing extremely exact and complex simulations. This method subdivides a complex structure into simpler parts (e.g., tetrahedral elements and nodes) accurately meshing the entire original structure. The major advantage of this method is that it allows for the individual analysis of the whole structure or only parts of it, which cannot be achieved otherwise. Thus, an FEA analysis in the periodontium and tooth allows measurements of stress distribution in small complex structures such as dental pulp, apical NVB and periodontal ligaments that is impossible in vivo [1]. Nevertheless, a validation of in vitro results must be performed whenever possible [1,2,3,4,14]. Due to the lack of studies about the apical NVB and dental pulp, a correlation with well-studied apical the third of PDL (i.e., which anatomically includes the apical NVB and should be reproduced 3D as well) was needed and considered to be acceptable. A simple and correct validation method could be a simple correlation between the amount of stress displayed by the FEA analysis and the MHP [1,2,3,4,15,16,17,18,19,20,21,22,23,24,25,26,27]. There are multiple periodontal ligament FEA studies that ignored this validation method, employed an unsuitable failure criterion, reported bias results exceeding the MHP values, and contradicted clinical and in vivo experiments [1,2,3,4,12,13,14,16,17,22,23,24,25].

FEA studies of the periodontium, especially of the periodontal ligament, employed either maximum principal tensile stress S1, minimum principal stress S3 [21,22,23,24,25,27] (which are suited only for brittle materials), or hydrostatic pressure [12,13,28] (suited only for fluids), over tissues possessing a clear ductile resemblance, thus providing debatable quantitative results that are in disagreement with clinical data. The hydrostatic pressure studies [12,13,15,19,20,21,28] employed a failure criteria showing a manner of loading characterized by no cellular deformation [29], not suited for PDL, dental pulp and NVB, and providing quantitative results that contradict the existing data of biomechanical behavior of all these tissues (e.g., PDL reports of stress amounts expressed in TPa [12,13] and MPa [28] instead of KPa [2,3,4,15,16,17] like for MHP). Only a few studies [1,2,3,4,21,22,23,30] used the adequate failure criteria (i.e., Von Mises and Tresca) based on analyzed material type (i.e., the ductile resemblance of PDL, dental pulp and NVB) showing quantitative results lower than MHP and in agreement with clinical data. Moreover, when investigating a complex structure such as the periodontium, for each individual part, a correlated failure criteria with the type of analyzed material should be used. Due to difficulties in 3D modeling, no studies investigating the apical NVB and dental pulp in periodontal breakdown were found. Thus, the need for a comparative study on anatomically correct 3D models employing the mostly used failure criterions assessing the biomechanical behavior of NVB and dental pulp under orthodontic movements during periodontal breakdown is obvious.

Due to a multitude of stress factors with regard to dental pulp in orthodontic treatment, the scientific validation needs further study [31,32,33,34,35]. Even the amount of force used for clinical studies (0.15–6 N) and applied for an average of 4 weeks in different movements was reported to supply inconclusive evidence [31,32,33,34,35]. Nonetheless, all studies [31,32,33,34,35] agreed that previously traumatized teeth with a history of periodontal problems are more prone to ischemia and necrosis during orthodontic movements [31,32,33,34,35].

Little data are available related to a correlation among the employment of the adequate failure criterion material type-based, maximum physiological hydrostatic pressure, the small orthodontic force, periodontal breakdown-tissues loss rate, ischemic-necrotic and orthodontic resorptive risks [1,2,3,4,15,19,20,21]. Thus, the study aims to bring further information related to this subject by applying five FEA criteria (usually employed in dentistry) in the study of NVB and dental pulp, assessing the adequate ones, and enabling the clinician to have new data related to the tissues’ biomechanical behavior [28].

The main objective was to perform an FEA simulation over NVB and dental pulp using the five FEA criteria (VM, Tresca, Pressure, S1 and S3) in both intact and 8 mm reduced periodontium for five pure orthodontic movements (intrusion, extrusion, translation, tipping, and rotation) and a small force of 0.5 N. A second goal was to select the most suitable criteria for apical NVB and dental pulp based on the correlation with MHP and the apical third of PDL. The last aim was to find an optimal orthodontic load safe to be used for apical NVB and dental pulp at various levels of tissue loss.

## 2. Materials and Methods

This study is a part of a larger research project stepwise developed aiming to investigate the periodontium and tooth during a periodontal breakdown process [1,2,3,4], continuing an earlier simulation conducted on the PDL [1]. The study’s clinical research protocol (158/2 April 2018) has been confirmed and approved by the Ethical Committee of the University of Medicine and Pharmacy Iuliu Hatieganu.

The research protocol required the existence of a non-inflamed periodontium, with variable levels of bone loss, orthodontic treatment request, and mandibular areas with no teeth loss. Only nine patients (mean age of 29.81 ± 1.45 years, six females) qualified for the study and gave the informed consent. Periodontal diagnosis for all nine cases was threated chronic periodontitis, various stages II/III, and grade B. The region of interest was the lower mandibular area with the two premolars and first molar, radiologically investigated using the CBCT (cone beam computed tomography), with a voxel size of 0.075 mm (CBCT, ProMax 3DS-Planmeca, Helsinki, Finland).

For the 3D reconstruction, CBCT-based AMIRA 5.4.0 (AMIRA, version 5.4.0, Visage Imaging Inc. Andover, MA, USA) software was employed. In order to ensure the anatomical correctness as much as possible, the manual reconstruction process based on the Hounsfield’s gray values (i.e., grays scale correlated with degree of x-ray attenuation by the tissues) was conducted by a single experienced clinician investigator. Each periodontium’s part (PDL with 0.15–0.225 mm thickness, cortical and trabecular bone, apical NVB), tooth (enamel, dentin, cement, dental pulp—for the second premolar) and bracket was manually reconstructed from each CBCT slice. The manual reconstruction process, despite being extremely time consuming and difficult, ensures a better anatomical correctness when compared with automated algorithm base reconstructions that meets difficulties when facing distinct types of tissues with a closely related radio-resemblance.

Following the reconstructive process, nine 3D models holding the second premolar and intact periodontium were obtained. In the next phase each of the models were subjected to a gradual horizontal reduction of 1 mm (up to 8 mm of tissue loss) of the bone and PDL. In total our simulations were performed over eighty-one models (Figure 1). The original nine reconstructed 3D models with no bone loss had 5.06–6.05 million C3D4 tetrahedral elements and 0.96–1.07 million nodes, while the global element size was 0.08–0.116 mm. As in any reconstructions, we found a reduced number of surface anomalies for each model, found in locations that do not interfere with stress concentrated areas that are quasi-continuous. It must be acknowledged that both types of software used contain built-in algorithms that prevent their use if too many surface anomalies are found and could potentially alter the results.

The analyses of apical NVB and dental pulp was conducted by employing five of the most used failure criteria in dentistry field: Von Mises, Tresca, Pressure, Maximum Principal and Minimum Principal in ABAQUS 6.11 (Dassault Systèmes, Vélizy-Villacoublay, France Manufacturer) FEA software version number. Five pure orthodontic movements were simulated: intrusion, extrusion, tipping, rotation and translation, and an amount of 0.5 N of low orthodontic force was applied at the bracket level.

The physical properties-boundary conditions (Table 1) implied homogeneity, isotropy, linear elasticity, perfectly bonded interfaces, and the encastred base of the model. The qualitative results (color coded projections) of the stress display in apical NVB and dental pulp is presented in Figure 2, Figure 3, Figure 4, Figure 5 and Figure 6. The quantitative results were correlated with the physiological MHP, quantitative results of the stress distribution in apical third of PDL, from the Part I of the study [1] (Table 2, Table 3 and Table 4). Based on these correlations, the ischemic, necrotic, and orthodontic root resorption risks were assessed.

## 3. Results

No significant differences between patients related to age, gender, periodontal status, or anatomy of the 3D models were found. The color-coded projections (i.e., qualitative results, Figure 2, Figure 3, Figure 4, Figure 5 and Figure 6, and Table 3) displayed the maximum stress in the apical NVB area in both intact and reduced periodontium for all five failure criteria. The quantitative results (i.e., Table 2 and Table 4 and Figure 2, Figure 3, Figure 4, Figure 5 and Figure 6) for both the intact and reduced periodontium revealed that apical NVB stress was significantly higher than the dental pulp stress, but significantly lower than the apical third of PDL areas. Moreover, all quantitative results were lower than the 16 KPa of MHP, up to 8 mm of bone loss (Table 2). The rotation movements seem to be the most invasive, while translation the least one among all five.

In the intact periodontium, an applied force of 0.5 N showed an average amount of Tresca shear stress 1.19 (apical NVB) and 1.06 (coronal pulp) times higher than VM overall stress. The VM and Tresca displayed the smallest amount of stress (i.e., in both pulpal and apical NVB) among the five investigated failure criteria (Table 2). The S3 Minimum principal compressive stress and hydrostatic Pressure stress displayed the highest amount of stress in apical NVB, lower than physiological MHP. At the coronal pulp level the highest amount of stress was showed by the Pressure failure criteria, followed by the S1 Maximum principal tensile stress and S3 compressive stress, as shown in Table 2. The translational movement, despite seeming to be the least invasive among all five, displayed (i.e., Figure 6) a Tresca and VM extremely visible coronal pulpal stress in the color-coded projection, lower than MHP. A similar stress was also visible for the rotational movement (Figure 4).

In reduced periodontium (1 to 8 mm of periodontal breakdown), the same amount of force produced amounts of Tresca average stress 1.17 (apical NVB) and 1.12 (coronal pulp) times higher than VM. The amounts of average stress produced by the two failure criteria remained much lower than the other three. The highest amounts of average stress were displayed by S3, S1 and Pressure criteria, lower than the 16 KPa of MHP. The same color-coded projection of visible coronal pulpal stress as in the intact periodontium are shown by translation and rotation (Figure 4 and Figure 6). Moreover, after 4 mm of bone loss, both movements displayed pulpal radicular stress.

Although producing average stresses much lower than the MHP, VM and Tresca seem to confirm both coronal and radicular visible stress for the translational and rotational movements up to 8 mm bone loss. The two failure criteria displayed identical stress areas (i.e., shear and overall) distribution on the color-coded projections. The S1, S3 and Pressure criteria showed visible amounts of stress in the apical NVB area (Figure 2, Figure 3, Figure 4, Figure 5 and Figure 6). The S3 showed a visible radicular pulp compressive stress after 4 mm of bone loss for rotation and translation to a smaller extent than with VM and Tresca.

The correlation between stress increase and bone loss was present in all five movements for all failure criteria.

Based on the amounts of stress displayed in Table 2 and Table 3, and color-coded projections (Figure 2, Figure 3, Figure 4, Figure 5 and Figure 6), it can be assumed that 0.5 N of orthodontic force is free of any ischemic, necrotic, and orthodontic root resorption risks up to 8 mm of bone loss. A clear difference between the accuracy of failure criteria based solely on the correlation between FEA quantitative results and MHP could not be proved (Table 2). However, a clear correlation between apical NVB, dental pulp, MHP, and PDL was found, and the difference between the accuracy of the criteria is clearly visible (Table 2 and Table 4).

## 4. Discussion

The here in comparative analysis is part of a larger research stepwise conducted [1,2,3,4] (i.e., study of intact and reduced periodontium under orthodontic forces) assessing the adequacy of different failure criteria for obtaining accurate results and being the single study of this type. This research simulated a gradual horizontal periodontal breakdown up to 8 mm of loss, conducted in 81 3D models (i.e., up to 6.05 million elements and 1.06 million nodes).

When investigating the current research flow, little data was available regarding the apical NVB and dental pulp biomechanical behavior in the intact periodontium, while for the reduced periodontium no studies were found. In orthodontic movements, the entire periodontium and tooth components are involved, thus for keeping the accuracy of results, correlations between different components (apical NVB—dental pulp—periodontal ligament) could and must be performed. Due to a lack of specialized studies analyzing the apical NVB and the need for validation of results [14], a correlation with apical PDL (i.e., that anatomically includes the NVB) studies was needed and considered acceptable (Table 4) [1,2,3,12,13,15,16,17,19,20,21,22,23,24,25,28,30].

Our results showed that in both intact and 8 mm reduced periodontium, 0.5 N of force is tolerable (i.e., lower than physiological MHP) for apical NVB and dental pulp in case of all five movements and failure criteria, thus excluding any ischemic, necrotic, and orthodontic root resorption risks. As expected, the apical NVB amounts of stress were significantly higher than the dental pulp’s exhibited stress, but lower than the apical third PDL stresses. These results agree with earlier reports [1,4] (Von Mises failure criteria in 0–8 mm bone loss, apical NVB and dental pulp, 0.6–1.2 N of orthodontic force, Table 4), other studies [1,2,3,4,5,6,7,8,9,10], and Proffit’s reports [11].

FEA is acknowledged as a correct research method allowing a multitude of individual simulations suppling both qualitative and quantitative data on each part of the analyzed structure [1,2,3,4,14,21,26]. However, three main requirements must be fulfilled: an adequate failure criterion (i.e., proper to the type of material from which the structure under investigation is made), anatomical accuracy of the structure, and boundary conditions (i.e., the structure’s physical properties) [1,2,3,4,14,21,26]. Due to its high accuracy, FEA is widely used to analyze and evaluate components, system strengths, and mechanical behavior under diverse types of environmental conditions in structural engineering, aerospace, the automotive industry, etc. However, in the study of PDL and tooth components, due to various reports of the same issue (e.g., optimal force in stressed PDL) but with variable different results (i.e., discrepancies between quantitative reports and clinical data, and lack of correlation with MHP [1,2,3,12,13,15,16,17,19,20,21,22,23,24,25,28,30]—Table 4), the FEA method is still regarded with care, while its results need direct and/or indirect validation. Most of the earlier studies with a focus on PDL partially acknowledged the importance of correct boundary conditions and anatomical accuracy requirements [1,2,3,4,5,6,7,8,9,10,12,13,14,15,16,17,18,19,20,21,22,23,24,25,26,27]. However, the first requirement mainly related to the yielding of different types of materials (complex and difficult problems even in the engineering field), has been completely overlooked [1,2,3,4,26], while the accuracy of most of the analyzed 3D models relayed mostly on simplified less accurate anatomical structures [12,13,15,16,17,19,20,21,22,24,25,27]. If in the engineering field the analyzed structure is usually made of a limited number of materials, in medicine and dentistry an anatomical structure could have multiple diverse types of materials that need diverse types of yielding methods of analysis (i.e., failure criteria). Thus, in structures of the periodontium and the tooth, the dental pulp, NVB and periodontal ligament more resemble ductile materials, while enamel, dentin, cement, bracket, cortical and trabecular bone are more similar to the brittle materials [1,2,3,4,26]. Besides their similarities, the main difference between ductile and brittle is related to their biomechanical behavior and their ability to deform under different loads and stresses. Thus, under a tensile load the material is subjected to elastic deformation shortly followed by the plastic deformation until the fracture and destruction occurs. The elastic deformation is temporary, while the plastic one is permanent. The ductility is the ability of a material to show plastic deformation before fracture (e.g., steel, rubber, plastics), while brittle materials (e.g., stone, ceramic, ice, cast iron) show negligible plastic deformations before fracture. The correct failure criteria for ductile materials are Von Mises and Tresca, for fluids it is hydrostatic Pressure, while for brittle materials they are Maximum S1 and Minimum S3 principal stresses. Unlike the engineering field, where the problems are far less complicated (i.e., fewer variables and unknowns, less complicated structures, and more easily quantifiable issues), when employing the FEA analysis (i.e., a mathematical algorithm-based method of analysis) in living tissues, there are usually variations of the quantitative results. Thus, due to variability in anatomy and physical properties of the tissues composing the periodontium and tooth, the assessment of the color-coded projections could provide variable quantitative results from one analysis to another (e.g., Geramy et al. [24,25], Toms et al. [23], Wu et al. [15,19,20], Moga et al. [1,2,3,4] reports—Table 4). However, each analysis could provide a certain pattern of biomechanical behavior of the analyzed structure that should remain unchanged for enabling the accuracy and validity of conclusions and therefore should agree with known clinical data. This stepwise research project implied multiple analysis on the same models with different aims, while few other studies performed similar simulations. As expected, there are differences between the quantitative stresses (Table 4), and due to continuous refinement of the FEA models (i.e., continuous surface smoothing), there are changes in the area of the applied orthodontic force (i.e., small changes of the surface and amount of loads), and assessment of the color-coded projections (i.e., different color-colored areas of stress display). Nonetheless, the pattern of biomechanical behavior and stress display is retained.

Table 4 displays the apical and cervical stress mostly in intact PDL when different failure criteria are employed. Here, the simulation employed all five criteria and compared the results with other studies [1,2,3,12,13,15,16,17,19,20,21,22,23,24,25,28,30] for investigating if significant differences were present. The reported quantitative values are comparable for studies using the same criterion (with small exceptions due to anatomical models and boundary conditions). However, the differences between the results of studies employing different failure criteria are significant. The closest is Von Mises and Tresca, which also provide results comparable with clinical data. The S1, S3 and hydrostatic pressure supply comparable quantitative reports, but higher than VM and Tresca, exceeding the 16 KPa of MHP (indirect validation criteria) even for no bone loss. Moreover, the hydrostatic pressure criteria studies (Wu et al. [15,19,20]) provided variable optimal amounts of force for intact PDL (i.e., 0.28–3.31 N) for different teeth (canine, premolar, lateral incisive), but with significant differences for the same tooth between similar reports (e.g., for canine: 1.7–2.1 N [15] and 3.31 N [19] for rotation and 0.38–0.4 N [15] and 2.3–2.6 N [20] for extrusion). For the premolar with no bone loss, the reported optimal rotational force was 2.8–2.9 N [19] opposingly to 0.5–0.6 N for intact PDL and 0.1–0.2 N 4–8 mm bone loss in present study.

The periodontal breakdown process was far less studied due to many difficulties related to the process (i.e., complexity and variability of the involved tissues). Thus, it must be admitted that a clinical study of the orthodontic movements, especially in periodontal breakdown, is impossible to be completed. The simulations performed in FEA analysis are of pure movements, despite the clinical ones being an association of movements.

Due to modeling difficulties, the FEA studies often only assessed only the intact PDL part using different failure criteria. However, the main issues observed in previous FEA studies of PDL [12,13,15,16,17,19,20,21,22,23,24,25,27,28,30] are related to the omission of at least one of the three main requirements for a correct conducted finite analysis: first—the criterion selection based on analyzed type of material, second—the correctness of the model and third—physical properties/boundary conditions as close as possible to the clinical ones (Table 4). Thus, the main issues found in these PDL studies are related to the use of improper failure criteria S1/S3 (correct for brittle materials) and hydrostatic pressure (correct for liquids), automated software simplified models, and lack of correlation between the quantitative reports with MHP (that is mandatory for validation [1,2,3,4]).

In the first part of the study [1] that analyzed the PDL’s biomechanical behavior under the same five failure criterions, we found that Tresca and Von Mises (correct for ductile resemblance materials) supply quantitative reports lower than physiological MHP (Table 4). Since apical NVB and dental pulp have the same ductile resemblance, they could and should be analyzed according to the same conditions for supplying the same reliability of results.

The reports of the first part of study [1] (using Tresca and Von Mises) concluded that the apical third of the periodontal ligament (i.e., especially reconstructed with apical NVB, and dental pulp) showed quantitative results lower than the 16 KPa of MHP up to 8 mm of periodontal breakdown (Table 4), which is in agreement with our findings. However, in reduced periodontium, at the cervical third the stresses exceeded by the physiological MHP, seemed to be prone to circulatory disturbances and ischemic/necrotic risks, thus the need to reduce the amount of applied orthodontic force to 0.1–0.2 N for rotation, 0.15–0.3 N for translation, and 0.2–0.4 N for tipping in 4–8 mm periodontal breakdown [1,2]. These observations could lead to the conclusion that in up to 8 mm tissue loss a force of 0.5 N has little to no effect on apical NVB, dental pulp and the apical third of PDL. This approach agrees with clinical studies of orthodontic movement effects over the dental pulp [31,32,33,34,35] (i.e., forces of 0.15–1.5 N seemed to produce only circulatory disturbances and limited histological changes, with little evidence of severe ischemia and necrosis). Nonetheless, the same studies [31,32,33,34,35] reported periodontal injury to display an elevated risk of ischemia, necrosis, and resorption, suggesting the need of using lower amounts of force, in agreement with our observations. The same amount of force might have a significant ischemic effect over the cervical third of PDL that could lead to further tissue loss, which agrees with other reports [12,13,16,17,24,25,28,30]. In both this and earlier studies of the rotation movement seems to be the most invasive. For apical NVB and dental pulp, the translation seems to be the least invasive, while in the same conditions, PDL (both apically and cervically) seems to be the second most invasive after rotation. This biomechanical behavior could be because PDL takes up and dissipates most of the stress of orthodontic movement, which is in agreement with the clinical data. These observations emphasize the need to correlate data between the biomechanical behavior of the most susceptible tissues to ischemic risks when performing orthodontic treatment in reduced periodontium and the importance of precaution [1,2,3,4] versus time reduction [21]. Moreover, in the first part of the study [1] PDL’s biomechanical behavior under S1, S3 and Pressure failure criteria (under 0.5 N of applied force), has met apically quantitative stress significantly higher than the MHP physiological value in both intact and reduced periodontium (Table 4). The cervical third stress was also significantly higher than the apical third stress. Based on the correlation between the amount of displayed stress-MHP [1,2,3,4,15,16,17,18,19,20,21] and the type of analyzed material assessment (brittle vs. ductile), a clear difference between the failure criteria suited for PDL analysis was found and then correlated with clinical data (i.e., only VM and Tresca found to be adequate and correct) [1]. However, in this study, due to the insignificant amount of stress displayed by the apical NVB and dental pulp, the same correlation between the amount of displayed stress and MHP was not found to be significantly decisive. Thus, the only available method for making the difference between criteria and of indirect validation is the association between the type of analyzed material assessment (PDL, dental pulp and NVB having all a ductile resemblance) [1,2,3,4,26] and PDL’s biomechanical behavior already investigated [1]. Relying on these aspects, we conclude that Von Mises and Tresca failure criteria seems to be more adequate and correct for apical NVB, and dental pulp’s FEA analysis is correct.

The physical properties also play a significant role in the correctness of an FEA analysis. Most of the earlier studies, including ours, employed isotropy, homogeneity, and linear elasticity, which are correct from the mechanical point of view when the tissues are subjected to small loads. Nevertheless, the main difference when assessing the quantitative reports came from the wrong choice of the material type-based criterion [1,2,3,4,26]. The hydrostatic stress implies no shear deformation, thus it is not applicable in tissue that suffer from shear deformation [1,29], implying consequences such as ischemia and necrosis. The Maximum and Minimum principal stress implies deformation, but with mathematical algorithm patterns for brittle materials [1,2,3,4,26], where the plastic deformation is negligible before fracture (e.g., concrete, and ceramic). Thus, only Von Mises and Tresca use a mathematical algorithm that describes the plastic deformation that occurs before fracture/yielding in living tissues such as PDL, apical NVB and dental pulp [1]. The main difference between them is that Tresca is designed to be more restrictive and conservative than Von Mises, and to allow a non-smooth behavior and a combination between a ductile flow and brittle fracture way [1,2,3,4,26].

From the biomechanical point of view, small loads considered to be approximately 1 N of force produce the same results no matter if they are employed in isotropy, homogeneity, or linear elasticity or in anisotropy, non-homogeneity, and non-linear elasticity [1,2,3,4,12,13,22,24,25,27]. Thus, due to ease in equations, the employment of isotropy, homogeneity, and linear elasticity is massively preferred in the analysis of PDL. Nonetheless, due to lack of studies on apical NVB and dental pulp, a direct comparative analysis could not be done (e.g., clinical studies investigated only whether a certain amount of force produced regressive changes in the dental pulp [31,32,33,34,35]). However, based on the data herein and from the first part of the study [1], it could be assumed that apical NVB and dental pulp could follow the same mechanical pattern as PDL, thus the same physical properties/boundary conditions could be applied. In an earlier series of studies [15,19,20] the Ogden hyper-elastic behavioral modality was used to describe PDL. Nevertheless, this model was especially designed for hyperplastic solids such as rubber (charged with a high number of strains), while human tissues do not follow the same pattern. Moreover, related to the linear versus non-linear behavior, Hemath et al. (2015) [16,17] (i.e., rising issues related to the incorrect use of criterion) reported a 10–30% difference between the quantitative results for S1 and S3 criteria in PDL (non-linear being higher than linear). However, even if this percentage were correct, the pattern of biomechanical behavior is retained, and the conclusions are still correct.

Our study analyzed pure movements which in clinical reality never happen, thus, the quantitative reports for the most stress areas, like apical NVB, could be even smaller. More finite element studies are needed to further analyze the correlation among different failure criteria, optimal force, material type, bone loss, MHP and ischemic/necrotic/resorptive risks.

## 5. Conclusions

The present findings show that:0.5 N of the orthodontic force is safe of any ischemic and necrotic risks up to 8 mm of periodontal breakdown, with rotation the most invasive (translation the least), and apical NVB more prone to ischemia than dental pulp. However, limited ischemic areas are also visible in the coronal and radicular pulp for translational and rotational movements.A clear difference between failure criteria when analyzing the dental pulp and NVB cannot be proved based only on the correlation between the FEA quantitative results and MHP. Nonetheless, the difference could rely on the type of analyzed material (i.e., ductile vs. brittle, with dental pulp, apical NVB and periodontal ligament having ductile resemblance), thus the choice should consider this issue. Tresca and VM’s failure criteria (adequate for the analysis of ductile materials) supplied comparable results and the lowest amount of stress.To obtain a clear image of the biomechanical behavior of tissues under orthodontic movements in a reduced periodontium, correlations with the other components of periodontium, such as PDL, should be considered.

## 6. Practitioner Points

Since previously traumatized teeth with periodontal injury are prone to ischemic and necrotic pulpal risks during orthodontic treatment, the amount of force should be reduced compared with those non-traumatized. Thus, up to 8 mm of periodontal breakdown 0.5 N of orthodontic force is safe to be used in all orthodontic movements with no ischemic and necrotic risks for apical NVB and dental pulp. The rotational movements seem to be the most invasive for apical NVB, dental pulp and PDL. Nonetheless, in orthodontic movements the entire periodontium is involved, thus for obtaining a clear image of the biomechanical behavior of tissues in reduced periodontium, correlations with the other components of periodontium, such as PDL, should and must be considered.

## Figures and Tables

**Figure 1 ijerph-19-15635-f001:**
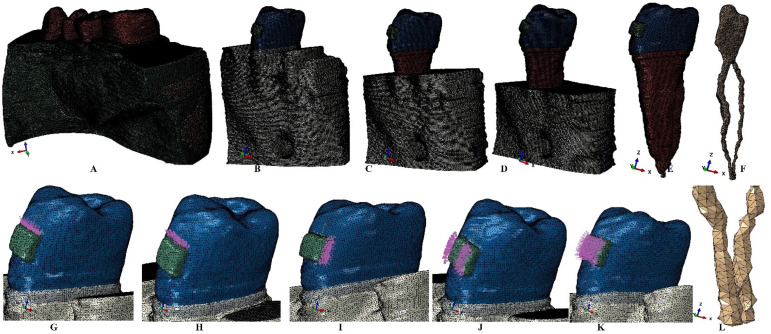
FEA mesh: (**A**)—one of the nine complete models (320 × 320 × 320 mm), (**B**)—2nd lower right premolar model (26 mm height) with intact periodontium, (**C**)—4 mm loss bone loss, (**D**)—8 mm bone loss, (**E**)—premolar (26 mm height) with NVB and bracket, (**F**)—dental pulp and NVB, orthodontic loads vectors: (**G**)—intrusion, (**H**)—extrusion, (**I**)—translation, (**J**)—rotation, (**K**)—tipping, (**L**)—NVB and apical third dental pulp.

**Figure 2 ijerph-19-15635-f002:**
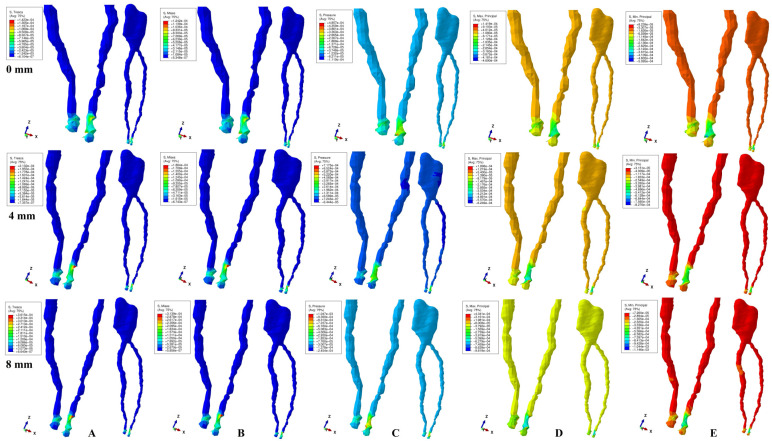
Stress distribution produced by intrusion movement in the apical NVB and dental pulp of one of the nine models (intact, 4 mm and 8 mm reduced periodontium, vestibular-mesial view) in MPa: (**A**)—Tresca, (**B**)—Von Mises, (**C**)—Pressure, (**D**)—Maximum principal, (**E**)—Minimum principal.

**Figure 3 ijerph-19-15635-f003:**
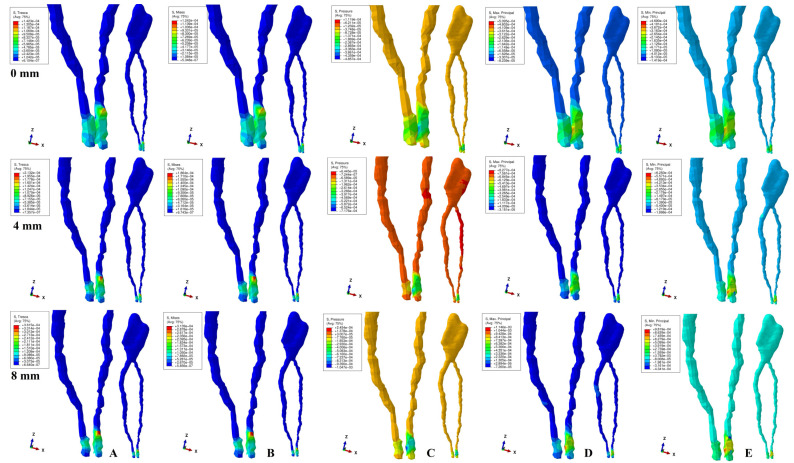
Stress distribution produced by extrusion movement in the apical NVB and dental pulp of one of the nine models (intact, 4 mm and 8 mm reduced periodontium, vestibular-mesial view) in MPa: (**A**)—Tresca, (**B**)—Von Mises, (**C**)—Pressure, (**D**)—Maximum principal, (**E**)—Minimum principal.

**Figure 4 ijerph-19-15635-f004:**
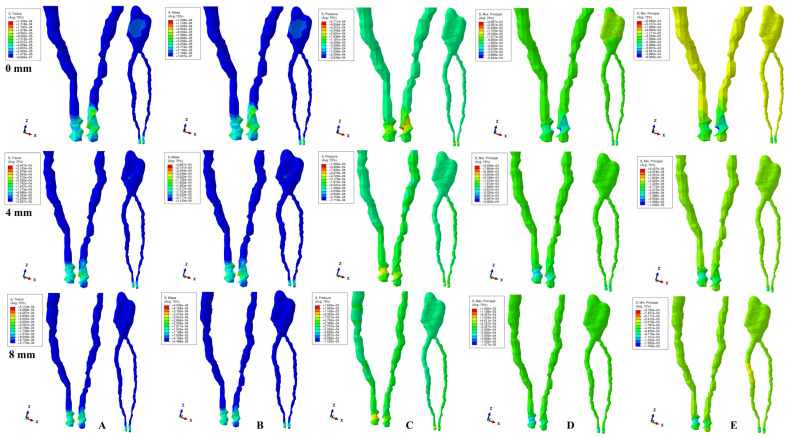
Stress distribution produced by rotation movement in the apical NVB and dental pulp of one of the nine models (intact, 4 mm and 8 mm reduced periodontium, vestibular-mesial view) in MPa: (**A**)—Tresca, (**B**)—Von Mises, (**C**)—Pressure, (**D**)—Maximum principal, (**E**)—Minimum principal.

**Figure 5 ijerph-19-15635-f005:**
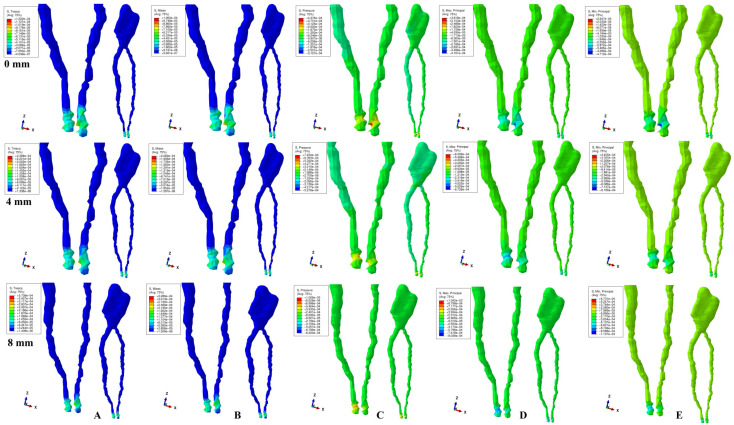
Stress distribution produced by tipping movement in the apical NVB and dental pulp of one of the nine models (intact, 4 mm and 8 mm reduced periodontium, vestibular-mesial view) in MPa: (**A**)—Tresca, (**B**)—Von Mises, (**C**)—Pressure, (**D**)—Maximum principal, (**E**)—Minimum principal.

**Figure 6 ijerph-19-15635-f006:**
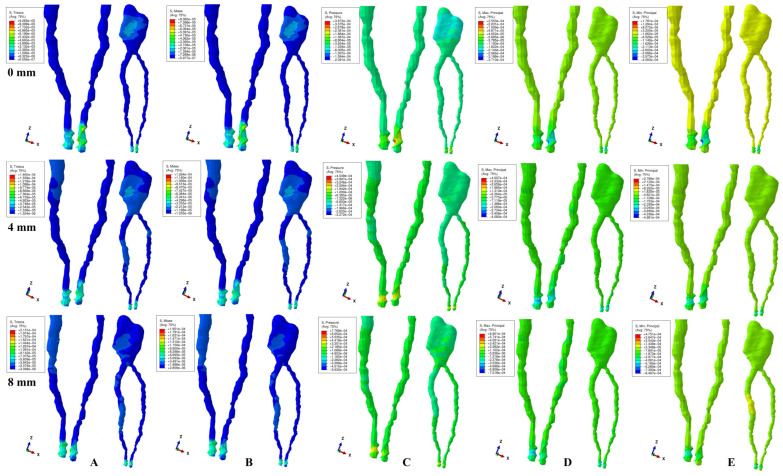
Stress distribution produced by translation movement in the apical NVB and dental pulp of one of the nine models (intact, 4 mm and 8 mm reduced periodontium, vestibular-mesial view) in MPa: (**A**)—Tresca, (**B**)—Von Mises, (**C**)—Pressure, (**D**)—Maximum principal, (**E**)—Minimum principal.

**Table 1 ijerph-19-15635-t001:** Elastic properties of materials.

Material	Young’s Modulus, E (GPa)	Poisson Ratio, ʋ	Refs.
Enamel	80	0.33	[1,2,3,4]
Dentin/Cementum	18.6	0.31	[1,2,3,4]
Pulp/apical NVB	0.0021	0.45	[1,2,3,4]
PDL	0.0667	0.49	[1,2,3,4]
Cortical bone	14.5	0.323	[1,2,3,4]
Trabecular bone	1.37	0.3	[1,2,3,4]
Bracket (Cr-Co)	218	0.33	[1,2,3,4]

**Table 2 ijerph-19-15635-t002:** Maximum stress average values (KPa) produced by orthodontic forces: NVB- apical neurovascular bundle; c-coronal pulp.

Resorption (mm)			0	1	2	3	4	5	6	7	8
Intrusion	Tresca	NVB	1.16	1.40	1.65	1.89	2.13	2.50	2.88	3.25	3.62
0.5 N		c	0.12	0.14	0.15	0.17	0.18	0.21	0.25	0.28	0.31
	VM	NVB	0.94	1.18	1.40	1.63	1.86	2.18	2.59	2.73	3.14
		c	0.11	0.12	0.14	0.15	0.16	0.19	0.22	0.24	0.27
	Pressure	NVB	4.36	4.94	5.45	5.98	6.52	7.52	8.50	9.48	10.47
		c	0.37	0.44	0.52	0.59	0.66	0.97	1.29	1.59	1.90
	S1	NVB	−3.70	−4.27	−4.83	−5.40	−5.96	−6.64	−7.33	−8.01	−8.69
		c	0.41	0.45	0.48	0.52	0.55	0.64	0.73	0.82	0.91
	S3	NVB	−4.60	−4.98	−5.37	−5.75	−6.13	−6.96	−7.78	−8.61	−9.43
		c	0.33	0.34	0.34	0.35	0.35	0.45	0.54	0.63	0.73
Extrusion	Tresca	NVB	1.20	1.40	1.65	1.89	2.13	2.50	2.88	3.25	3.62
0.5 N		c	0.12	0.14	0.15	0.17	0.18	0.21	0.25	0.28	0.31
	VM	NVB	0.94	1.18	1.40	1.63	1.86	2.18	2.59	2.73	3.14
		c	0.11	0.12	0.14	0.15	0.16	0.19	0.22	0.24	0.27
	Pressure	NVB	−4.36	−4.94	−5.45	−5.98	−6.52	−7.52	−8.50	−9.48	−10.47
		c	−0.37	−0.44	−0.52	−0.59	−0.66	−0.97	−1.29	−1.59	−1.90
	S1	NVB	4.21	4.97	5.74	6.50	7.26	7.82	8.38	8.93	9.49
		c	−0.83	−0.92	−1.02	−1.12	1.19	1.24	1.29	1.34	1.39
	S3	NVB	4.69	5.09	5.48	5.86	6.25	7.14	8.04	8.93	9.82
		c	−0.40	−0.44	−0.47	−0.51	−0.54	−0.61	−0.68	−0.74	−0.80
Translation	Tresca	NVB	0.93	1.06	1.20	1.33	1.46	1.63	1.81	1.98	2.15
0.5 N		c	0.16	0.18	0.21	0.23	0.25	0.26	0.28	0.29	0.30
	VM	NVB	0.81	0.92	1.04	1.15	1.26	1.43	1.61	1.78	1.95
		c	0.21	0.22	0.23	0.24	0.25	0.25	0.26	0.26	0.26
	Pressure	NVB	3.87	4.05	4.24	4.41	4.59	5.39	6.19	6.98	7.77
		c	−0.56	−0.58	−0.60	−0.62	0.64	0.75	0.86	0.96	1.07
	S1	NVB	−3.29	−3.51	−3.74	−3.98	−4.18	−4.89	−5.61	−6.32	−7.03
		c	0.41	0.47	0.53	0.58	0.64	0.71	0.78	0.84	0.91
	S3	NVB	−4.06	−4.29	−4.52	−4.75	−4.98	−5.87	−6.75	−7.63	−8.50
		c	−0.17	−0.17	−0.18	−0.18	0.18	0.22	0.26	2.93	0.33
Rotation	Tresca	NVB	1.43	1.51	2.01	2.52	3.45	3.87	4.28	4.70	5.11
0.5 N		c	0.25	0.27	0.29	0.30	0.32	0.36	0.40	0.44	0.47
	VM	NVB	1.25	1.62	1.98	2.34	2.71	2.89	3.08	3.26	3.44
		c	0.22	0.24	0.25	0.27	0.28	0.32	0.37	0.41	0.45
	Pressure	NVB	5.71	6.64	7.55	8.47	9.39	10.55	11.72	12.88	14.03
		c	2.69	2.82	2.94	3.06	3.17	3.55	3.94	4.32	4.70
	S1	NVB	−5.55	−6.63	−7.78	−8.89	−10.00	−11.20	−12.40	−13.59	−14.79
		c	0.94	1.47	2.00	2.52	3.05	3.46	3.87	4.28	4.69
	S3	NVB	−6.03	−7.14	−8.25	−9.35	−10.46	−11.75	−13.04	−14.34	−15.62
		c	0.63	0.78	0.94	1.09	−1.23	−1.37	−1.52	−1.66	−1.80
Tipping	Tresca	NVB	1.22	1.52	1.81	2.11	2.40	2.74	3.07	3.41	3.74
0.5 N		c	0.21	0.22	0.23	0.23	0.24	0.26	0.29	0.30	0.32
	VM	NVB	1.06	1.32	1.57	1.83	2.08	2.39	2.70	3.00	3.30
		c	0.18	0.18	0.19	0.19	0.19	0.22	0.24	0.27	0.29
	Pressure	NVB	4.38	5.14	5.92	6.68	7.45	8.24	9.03	9.81	10.60
		c	1.88	1.95	2.02	2.08	2.15	2.21	2.28	2.34	2.40
	S1	NVB	−4.26	−4.67	−5.08	−5.49	−6.97	−7.61	−8.26	−8.90	−9.54
		c	−0.84	−1.16	−1.47	−1.79	2.10	2.56	3.03	3.49	3.95
	S3	NVB	−4.71	−5.32	−5.93	−6.53	−7.14	−8.12	−9.11	−10.09	−11.07
		c	−0.42	−0.50	−0.58	−0.66	−0.74	0.76	0.78	0.79	0.80

**Table 3 ijerph-19-15635-t003:** Color-coded stress projection in apical NVB and dental pulp for different failure criteria.

Resorption (mm)		Intact Periodontium	8 mm Reduced Periodontium
Intrusion	Tresca	NVB	NVB
0.5 N	Von Mises	NVB	NVB
	Pressure	NVB	NVB
S1	Max. Princ.	NVB	NVB
S3	Min. Princ.	NVB	NVB
Extrusion	Tresca	NVB	NVB
0.5 N	Von Mises	NVB	NVB
	Pressure	NVB	NVB
S1	Max. Princ.	NVB	NVB
S3	Min. Princ.	NVB	NVB
Translation	Tresca	NVB, C, r	NVB, c
0.5 N	Von Mises	NVB, C, r	NVB, c
	Pressure	NVB	NVB, c
S1	Max. Princ.	NVB	NVB
S3	Min. Princ.	NVB	NVB, r
Rotation	Tresca	NVB, c	NVB
0.5 N	Von Mises	NVB, c	NVB
	Pressure	NVB,	NVB
S1	Max. Princ.	NVB,	NVB
S3	Min. Princ.	NVB, c	NVB, r
Tipping	Tresca	NVB	NVB
0.5 N	Von Mises	NVB	NVB
	Pressure	NVB	NVB
S1	Max. Princ.	NVB	NVB
S3	Min. Princ.	NVB	NVB

r—radicular pulp lower intensity, c—coronal pulp lower intensity. NVB—higher intensity, C—coronal pulp-higher intensity.

**Table 4 ijerph-19-15635-t004:** Quantitative reports for apical NVB and PDL (a—apically, c—cervically).

FEA Criterion	Studies	Force, Movement, Stress, NVB and PDL
Von Mises	Moga et al. (2019) [4], lower premolar	0.2 N intrusion; 0.5 KPa NVB
	5.06–6.05 million elements; 0.96–1.07 million nodes	0.6 N extrusion; 1.5 KPa NVB
	No bone loss	1.2 N translation; 1.9 KPa NVB
		0.6 N rotation; 1.5 KPa NVB
		0.6 N tipping; 1.2 KPa NVB
	8 mm bone loss	0.2 N intrusion; 1.4 KPa NVB
		0.6 N extrusion; 4.1 KPa NVB
		1.2 N translation; 4.7 KPa NVB
		0.6 N rotation; 5.4 KPa NVB
		0.6 N tipping; 3.9 KPa NVB
	Toms et al. (2003) [23], lower premolar, 5205 nodes and 1674 elements	1 N extrusion; a-8 KPa; c-7.75 KPa
	No bone loss	
	Merdji et al. (2013) [30], lower molar, 557,974 elements	10 N intrusion; a-29.48 KPa
	No bone loss	3 N tipping; a-8.96 KPa
		3 N translation; a-6.78 KPa
	Shaw et al. (2004) [22], upper incisor, 20,582 nodes and 11,924 elements	? N extrusion-intrusion a-2 KPa
	No bone loss	? N tipping a-1 KPa
	Roscoe et al. (2021) [21], premolar, 1.67 million elements	0.25 N intrusion; a/c-1.1 KPa
	No bone loss	0.25 N tipping; a/c-2.9 KPa
	Moga et al. (2022) [2], lower premolar, 0.2–1.2 N	0.2 N intrusion; a-0.44 KPa, c-1.51 KPa
	5.06–6.05 million elements; 0.96–1.07 million nodes	0.6 N extrusion; a-1.33 KPa, c-5.18 KPa
	No bone loss	1.2 N translation; a-3.58 KPa, c-28.06 KPa
		0.6 N rotation; a-2.02 KPa, c-15.91 KPa
		0.6 N tipping; a-1.34 KPa, c-10.52 KPa
	8 mm bone loss	0.2 N intrusion; a-1.22 KPa, c-4.76 KPa
		0.6 N extrusion; a-5.42 KPa, c-21.39 KPa
		1.2 N translation; a-26.28 KPa, c-117 KPa
		0.6 N rotation; a-17.86 KPa, c-71.06 KPa
		0.6 N tipping; c-7.29 KPa, c-43.19 KPa
	Moga et al. (2022) [1], lower premolar, 0.5 N	Intrusion; a-2.17 KPa, c-4.32 KPa
	5.06–6.05 million elements; 0.96–1.07 million nodes	Extrusion, a-2.17 KPa, c-4.85 KPa
	No bone loss	Translation; a-1.49 KPa, c-14.59 KPa
		Rotation; a-1.68 KPa, c-14.80 KPa
		Tipping, a-1.12 KPa, c-9.85 KPa
	8 mm bone loss	Intrusion; a-6 KPa, c-11.19 KPa
		Extrusion; a-6 KPa, c-16.34 KPa
		Translation; a-5.49KPa, c-54.44 KPa
		Rotation; a-7.49 KPa, c-66.46 KPa
		Tipping; a-6 KPa, c-33 KPa
Tresca	Moga et al. (2022) [1], lower premolar, 0.5 N	Intrusion; a-2.50 KPa, c-4.97 KPa
	5.06–6.05 million elements; 0.96–1.07 million nodes	Extrusion; a-2.50 KPa, c-5.59 KPa
	No bone loss	Translation; a-1.68 KPa, c-16.37 KPa
		Rotation, a-1.94 KPa, c-17.17 KPa
		Tipping; a-1.29 KPa, c-11.36 KPa
	8 mm bone loss	Intrusion; a-6.85 KPa, c-13.66 KPa
		Extrusion, a-6.87 KPa, c-18.75 KPa
		Translation; a-6.31 KPa, c-62.57 KPa
		Rotation; a-8.23 KPa, c-73 KPa
		Tipping; a-6.89 KPa, c-37.42 KPa
S1–S3	Toms et al. (2003) [23], lower premolar, 5205 nodes; 1674 elements, 1 N	Extrusion; S1: a-36 KPa, c-(−2.69 KPa)
	No bone loss	Extrusion; S3: a-28.49 KPa, c-(−11.6 KPa)
	Moga et al. (2021) [3], lower premolar 0.2–1.2 N	0.2 N intrusion; S3: a-(−1.74 KPa), c-(−1.74 KPa)
	5.06–6.05 million elements; 0.96–1.07 million nodes	0.6 N extrusion; S3: a-14.1 KPa, c-27.99 KPa
	No bone loss	1.2 N translation; S3: a-(−97.79 KPa), c-93 KPa
		0.6 N rotation; S3: a-(−56.27 KPa), c-68.07 KPa
		0.6 N tipping; S3: a-(−18.53 KPa), c-28.89 KPa
	8 mm bone loss	0.2 N intrusion; S3: a-(−21.26 KPa), c-(−8.8 KPa)
		0.6 N extrusion; S3: a-64.15 KPa, c-82.83 KPa
		1.2 N translation; S3: a-(−292 KPa), c-260 KPa
		0.6 N rotation; S3: a-(−290 KPa), c-170 KPa
		0.6 N tipping; S3: a-(−109 KPa), c-(−109 KPa)
	Geramy et al. (2004) [24], upper central incisor	Tipping; S1: a-78 KPa, c-(−23.6 KPa)
	378,884 nodes, 32,768 elements, 1 N	Tipping; S3: a-(–74 KPa), c-(−28 KPa)
	No bone loss	
	8 mm bone loss	Tipping; S1: a-881 KPa, c-(−395 KPa)
		Tipping; S3: a-740 KPa, c-(−491 KPa)
	Geramy et al. (2002) [25], upper central incisor	Tipping; S1: a-(−37 KPa), c-55 KPa
	726 nodes, 475 elements, 1 N	Tipping; S3: a-39 KPa, c-(−75 KPa)
	No bone loss	
		Intrusion; S1: a-26 KPa, c-(−9 KPa)
		Intrusion; S3: a-(−29 KPa), c-(−12 KPa)
	8 mm bone loss	Tipping; S1: a-(−440 KPa), c-(−288 KPa)
		Tipping; S3: a-(−475 KPa), c-300 KPa
		Intrusion; S1: a-(–43 KPa), c-19 KPa
		Intrusion; S3: a-(–47 KPa), c-(−23 KPa)
	Hemanth et al. (2015) [16,17], upper central incisor	0.2 N intrusion; S1: c-1 KPa
	239,666 nodes, 148,097 elements.	1 N tipping; S1: a-(−16.4 KPa)
	No bone loss	
		0.2 N intrusion; S3: a-(−13.37 KPa)
		1 N tipping; S3: a-16.4 KPa
	Roscoe et al. (2021) [21], premolar, 1.67 million elements	0.25 N intrusion; a/c-5.3 KPa
	No bone loss	0.25 N tipping; a/c-7.3 KPa
Pressure	Moga et al. (2022) [1], lower premolar, 0.5 N	Intrusion; a-(−13.68 KPa), c-18.86 KPa
	5.06–6.05 million elements; 0.96–1.07 million nodes	Extrusion; a-13.68 KPa, c-19.1 KPa
	No bone loss	Translation; a-(−28.21 KPa), c-(−79.1 KPa)
		Rotation, a-(−34.3 KPa), c-(−85.3 KPa)
		Tipping; a-13.75 KPa, c-33 KPa
	8 mm bone loss	Intrusion; a-(−30 KPa), c-60.8 KPa
		Extrusion, a-30 KPa, c-60.3 KPa
		Translation; a-103.8 KPa, c-(−129 KPa)
		Rotation; a-97.6 KPa, c-(−162 KPa)
		Tipping; a-71.3 KPa, c-129 KPa
	Hohmann et al. (2009) [13], upper premolar, 0.5 N	Intrusion, a-(4.7 KPa–9.95 TPa), c-4.7 KPa
	PDL 195,881 elements, tooth 711,114 elements	
	No bone loss	
	Hohmann et al. (2007) [12], upper premolar 3 N	Tipping; a-38.84 KPa, c-(−68 KPa)
	PDL 152,776 elements, tooth 56,454 elements	
	No bone loss	
	Wu et al. (2018) [15], upper canine	optimal force: tipping 0.28–0.44 N, translation 1.1–1.37 N
	PDL 1263, elements, tooth 1928 elements	rotation 1.7–2.1 N, extrusion 0.38–0.4 N
	No bone loss	
	Wu et al. (2021) [19], lower incisor, canine, premolar	optimal force: rotation 2.2–2.3 N, 3–3.1 N, 2.8–2.9 N
	PDL 3032, 3416, 3851 elements, bone 5692 elements	
	No bone loss	
	Wu et al. (2019) [20], upper canine	optimal force: intr. 0.8–0.9 N, extr. 2.3–2.6 N
	PDL 2272, elements, tooth 2101 elements	
	No bone loss	
	Roscoe et al. (2021) [21], premolar, 1.67 million elements, 0.25 N	Intrusion; a/c-(−4.7 KPa)
	No bone loss	Tipping; a/c-(−5.8 KPa)
	Zhong et al. (2019) [28], lower premolar, PDL 17,575 elements, 0.25 N	Tipping; a/c-(10–20 KPa)
	No bone loss	

NVB-apical neuro-vascular bundle, a-apical third PDL, c-cervical third PDL.

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
