# Peer review of "Assessment of the Best FEA Failure Criteria (Part II): Investigation of the Biomechanical Behavior of Dental Pulp and Apical-Neuro-Vascular Bundle in Intact and Reduced Periodontium"

_ijerph, 2022, doi:10.3390/ijerph192315635_

Round 1
Reviewer 1 Report
the manuscript fits well in the journal as it describes an important issue for dentistry and ortodontics. I suggest some minor revision in the manuscript content.
1) The introduction is way to long. Please summarize the first paragraphs in only one (Page 2 and 3); in the introduction it is not necessary to state all the limits of FEA applied in ortodontics.
Page 4 first paragraph. Please revise the english
Page 5 results. please specify which differences authors accounted: what about patient distribution, this is not reported.
it should be reported which clinical application the study refers (ortodontic classes, open/closed bite.......). Orthodontic movement of periodontally damaged teeth is quite unusual. Please clarify.
Author Response
Corresponding author
Department of Cariology, Endodontics and Oral Pathology
School of Dental Medicine
University of Medicine and Pharmacy Iuliu Hatieganu
Dr. Pidchayanin Wongsarakit
Assigned Editor
International Journal of Environmental Research and Public Health
Special Issue - Advances of Digital Dentistry and Prosthodontics
November 18, 2022
Dear Dr. Pidchayanin Wongsarakit,
Thank you very much for your letter dated November 15, 2022, with the comments of the reviewers. We have now carefully considered the comments of the reviewers and amended the paper accordingly. All changes are highlighted in red throughout the manuscript and included also below.
Reply to Reviewer #1:
We agree and we thank the reviewer for his/her time and comments. Appropriate changes in the manuscript have by now been made. Please see below and in the manuscript.
Concern of the reviewer: The manuscript fits well in the journal as it describes an important issue for dentistry and ortodontics. I suggest some minor revision in the manuscript content.
- The introduction is way to long. Please summarize the first paragraphs in only one (Page 2, and 3);
- in the introduction it is not necessary to state all the limits of FEA applied in ortodontics.
Our response:
- We thank the reviewer for his/her concern and comments. We do hope that our changes are according to the reviewer‘s remarks.
Revised text: entire Introduction section
Concern of the reviewer: Page 4 first paragraph. Please revise the English
Our response:
- We thank the reviewer for his/her concern and comments. We do hope that our changes are according to the reviewer‘s remarks.
Revised text: Material and methos section, pg. 4 lines:154-168
“This study is a part of a larger research project stepwise developed aiming to investigate the periodontium and tooth during a periodontal breakdown process [1-4], continuing an earlier simulation conducted on the PDL [1]. The study’s clinical research protocol (158/02.04.2018) has been confirmed and approved by the Ethical Committee of the University of Medicine and Pharmacy Iuliu Hatieganu.
The research protocol required the existence of non-inflamed periodontium, with variable levels of bone loss, orthodontic treatment request, and mandibular areas with no teeth loss. Only nine patients (mean age of 29.81 ± 1.45 years, 6 females) qualified for the study and gave the informed consent. Periodontal diagnosis for all nine cases was threated chronic periodontitis, various stages II/III, and grade B. The region of interest was the lower mandibular area with the two premolars and first mormolar, which was radiologically investigated using the CBCT (cone beam computed tomography) with a voxel size of 0.075 mm (CBCT, ProMax 3DS-Planmeca, Finland).”
Concern of the reviewer: Page 5 results. please specify which differences authors accounted: what about patient distribution, this is not reported.
Our response:
- We thank the reviewer for his/her concern and comments. We do hope that our changes are according to the reviewer‘s remarks.
Revised text: Material and methos section, pg. 4 lines:162
“The research protocol required the existence of non-inflamed periodontium, with variable levels of bone loss, orthodontic treatment request, and mandibular areas with no teeth loss. Only nine patients (mean age of 29.81 ± 1.45 years, 6 females) qualified for the study and gave the informed consent.”
Concern of the reviewer: it should be reported which clinical application the study refers (ortodontic classes, open/closed bite.......). Orthodontic movement of periodontally damaged teeth is quite unusual. Please clarify.
Our response:
- We thank the reviewer for his/her concern and comments. We do hope that our changes are according to the reviewer‘s remarks.
Revised text: Practitioner Points section, pg. 17-18 lines: 507-515
“Since previously traumatized teeth with periodontal injury are prone to ischemic and necrotic pulpal risks during orthodontic treatment, the amount of force should be reduced compared with those non-traumatized. Thus, up to 8 mm of periodontal breakdown 0.5 N of orthodontic force is safe to be used in all orthodontic movements with no ischemic and necrotic risks for apical NVB and dental pulp. The rotational movements seem to be the most invasive for apical NVB, dental pulp and PDL. Nonetheless, in orthodontic movements the entire periodontium is involved, thus for obtaining a clear im-age of the biomechanical behavior of tissues in reduced periodontium, correlations with the other components of periodontium, such as PDL should and must be considered.”

Reviewer 2 Report
Dear authors, congratulations for the study. However I have some comments
Introduction
1. Excessive self-citation should be avoided. The first four bibliographic references in the introduction are the author's own. Without underestimating the research experience in this field reference should be made to other articles of interest in the introduction.
2. This paragraph should be referenced. “EA is acknowledged as a correct research method allowing a multitude of individual simulations suppling both qualitative and quantitative data on each part of the analyzed structure. However, for being correct when employed, three main requirements must be fulfilled: an adequate failure criterion (i.e., proper to the type of material from which the structure under investigation is made), anatomical accuracy of the structure and boundary conditions (i.e., structure’s physical properties). Due to its high accuracy FEA is widely used to analyze and evaluate components, systems strengths, and mechanical behavior under diverse types of environmental conditions in structural engineering, aerospace, car industry, etc. However, in medicine and dentistry field due to unexplained inconsistencies between quantitative results and clinical data and misuse, FEA is regarded with undeserved reserve and distrust. Most of the earlier studies with focus on PDL partially acknowledged the importance of correct boundary conditions and anatomical accuracy requirements.”
3. Try to avoid the first person plural in the text.
4. “However, in medicine and dentistry field due to unexplained inconsistencies between quantitative results and clinical data and misuse, FEA is regarded with undeserved reserve and distrust”. The authors make this claim in a generalised way by referring only to their own publications. It is not very correct, from a research ethics point of view, to include such phrases. If you want to evaluate the work of other professionals, I suggest that you directly carry out a comparative study (through meta-analysis) comparing the results of your studies with those of others.
Materials and Methods
1. “For the 3D reconstruction CBCT-based AMIRA 5.4.0 software was employed.” The product manufacturer and country of origin must be specified.
2. “ABAQUS 6.11 FEA software.” The product manufacturer and country of origin must be specified.
3. Figure 1. The upper images look excessively dark.
Results:
1. The values in figures 2 to 6 are not displayed correctly. It is suggested that the authors improve the image quality or reduce the number of figures to improve the visualisation.
2. Table IV is not directly related to the research results of this study and its analysis should be included in the discussion.
Discusión
1. At least 22 references are made in the discussion to articles written by the authors. It is recommended to discuss the results with other studies directly related to the results of this article.
3. As indicated above, include the information in table IV in the discussion.
4. It would be useful to include in the discussion references to published studies on orthodontic force damage and to compare them with the results obtained in order to relate the clinical performance of the orthodontist to the possible damage that may be caused by his or her treatment.
Conclusions
It is recommended that the number of conclusions be reduced by grouping conclusions related to the same topic under a single point.
Author Response
Corresponding author
Department of Cariology, Endodontics and Oral Pathology
School of Dental Medicine
University of Medicine and Pharmacy Iuliu Hatieganu
Dr. Pidchayanin Wongsarakit
Assigned Editor
International Journal of Environmental Research and Public Health
Special Issue - Advances of Digital Dentistry and Prosthodontics
November 18, 2022
Dear Dr. Pidchayanin Wongsarakit,
Thank you very much for your letter dated November 15, 2022, with the comments of the reviewers. We have now carefully considered the comments of the reviewers and amended the paper accordingly. All changes are highlighted in red throughout the manuscript and included also below.
Reply to Reviewer #2:
We agree and we thank the reviewer for his/her time and comments. Appropriate changes in the manuscript have by now been made. Please see below and in the manuscript.
Concern of the reviewer:
Dear authors, congratulations for the study. However, I have some comments
Introduction
- Excessive self-citation should be avoided. The first four bibliographic references in the introduction are the author's own. Without underestimating the research experience in this field reference should be made to other articles of interest in the introduction.
- This paragraph should be referenced. “EA is acknowledged as a correct research method allowing a multitude of individual simulations suppling both qualitative and quantitative data on each part of the analyzed structure. However, for being correct when employed, three main requirements must be fulfilled: an adequate failure criterion (i.e., proper to the type of material from which the structure under investigation is made), anatomical accuracy of the structure and boundary conditions (i.e., structure’s physical properties). Due to its high accuracy FEA is widely used to analyze and evaluate components, systems strengths, and mechanical behavior under diverse types of environmental conditions in structural engineering, aerospace, car industry, etc. However, in medicine and dentistry field due to unexplained inconsistencies between quantitative results and clinical data and misuse, FEA is regarded with undeserved reserve and distrust. Most of the earlier studies with focus on PDL partially acknowledged the importance of correct boundary conditions and anatomical accuracy requirements.”
- Try to avoid the first person plural in the text.
- “However, in medicine and dentistry field due to unexplained inconsistencies between quantitative results and clinical data and misuse, FEA is regarded with undeserved reserve and distrust”. The authors make this claim in a generalised way by referring only to their own publications. It is not very correct, from a research ethics point of view, to include such phrases. If you want to evaluate the work of other professionals, I suggest that you directly carry out a comparative study (through meta-analysis) comparing the results of your studies with those of others.
Our response:
- We thank the reviewer for his/her concern and comments. We do hope that our changes are according to the reviewer‘s remarks.
1.Revised text: entire Introduction section,
- Revised text: Discussion section, pg. 14 lines:310-323,
“FEA is acknowledged as a correct research method allowing a multitude of individual simulations suppling both qualitative and quantitative data on each part of the analyzed structure [1-4, 14, 21, 26]. However, for being correct when employed, three main requirements must be fulfilled: an adequate failure criterion (i.e., proper to the type of material from which the structure under investigation is made), anatomical accuracy of the structure and boundary conditions (i.e., structure’s physical properties) [1-4, 14, 21, 26]. Due to its high accuracy FEA is widely used to analyze and evaluate components, systems strengths, and mechanical behavior under diverse types of environmental conditions in structural engineering, aerospace, car industry, etc. However, in the study of PDL and tooth components, due various reports of the same issue (e.g., optimal force in stressed PDL) but with variable different results (i.e., discrepancies between quantitative reports and clinical data, and lack of correlation with MHP [1, 2, 3, 12, 13, 15-17, 19-25, 28, 30] - Table 4), FEA method is still regarded with care, while its results need direct and/or indirect validation.”
- Revised text: entire Introduction section,
- Revised text: Discussion section, pg. 14 lines:318-323,
“However, in the study of PDL and tooth components, due various reports of the same issue (e.g., optimal force in stressed PDL) but with variable different results (i.e., discrepancies between quantitative reports and clinical data, and lack of correlation with MHP [1, 2, 3, 12, 13, 15-17, 19-25, 28, 30] - Table 4), FEA method is still regarded with care, while its results need direct and/or indirect validation.”
Concern of the reviewer: Materials and Methods
- “For the 3D reconstruction CBCT-based AMIRA 5.4.0 software was employed.” The product manufacturer and country of origin must be specified.
- “ABAQUS 6.11 FEA software.” The product manufacturer and country of origin must be specified.
- Figure 1. The upper images look excessively dark.
Our response:
- We thank the reviewer for his/her concern and comments. We do hope that our changes are according to the reviewer‘s remarks.
1.Revised text: Material and Methods section, pg. 4, lines 167-170
“The region of interest was the lower mandibular area with the two premolars and first molar, radiologically investigated using the CBCT (cone beam computed tomography) with a voxel size of 0.075 mm (CBCT, ProMax 3DS-Planmeca, Finland).
For the 3D reconstruction CBCT-based AMIRA 5.4.0 (AMIRA, version 5.4.0, Visage Imaging Inc. 300 Brickstone Square, Suite 201 Andover, MA 01810, USA) software was employed.”
- Revised text: Material and Methods section, pg. 4, lines 194
“The analyses of NVB and dental pPulp was conducted by employing five of the most used failure criteria in dentistry field: Von Mises, Tresca, Pressure, Maximum Principal and Minimum Principal in ABAQUS 6.11 (Dassault Systèmes-France) FEA software”
- The word did not keep the accuracy of the figure. We tried to insert once more the high-resolution picture, but the high resolution was not kept. We already send the high-resolution files.
Concern of the reviewer: Results:
- The values in figures 2 to 6 are not displayed correctly. It is suggested that the authors improve the image quality or reduce the number of figures to improve the visualisation.
- Table IV is not directly related to the research results of this study and its analysis should be included in the discussion.
Our response:
- We thank the reviewer for his/her concern and comments. We do hope that our changes are according to the reviewer‘s remarks.
- The word did not keep the accuracy of the figure. We tried to insert once more the high-resolution picture, but the high resolution was not kept. We already send the high-resolution files.
- Revised text: Discussion section, pg. 15, lines 366-381
“Table 4 display the apical and cervical stress mostly in intact PDL when different failure criteria are employed. Here simulation employed all five criteria and compared the results with other studies [1, 2, 3, 12, 13, 15-17, 19-25, 28, 30] for investigating if significant differences are present. The reported quantitative values are comparable for studies using the same criterion (with small exceptions due to anatomical models and boundary conditions). However, the differences between the results of studies employing different failure criteria are significant. The closet is Von Mises and Tresca, which also provide results comparable with clinical data. The S1, S3 and hydrostatic pressure supply comparable quantitative reports, but higher than VM and Tresca, exceeding the 16 KPa of MHP (indirect validation criteria) even for no bone loss. Moreover, the hydrostatic pressure criteria studies (Wu et al. [15, 19, 20]) provided variable optimal amounts of force for intact PDL (i.e., 0.28-3.31 N) for different teeth (canine, premolar, lateral incisive), but with significant differences for the same tooth between similar reports (e.g., for canine: 1.7-2.1 N [15] and 3.31 N [19] for rotation and 0.38-0.4 N [15] and 2.3-2.6 N [20] for extrusion). For the premolar with no bone loss, the reported optimal rotational force was 2.8-2.9 N [19] opposingly to 0.5-0.6 N for intact PDL and 0.1-0.2 N 4-8 mm bone loss in present study.”
Concern of the reviewer: Discusión
- At least 22 references are made in the discussion to articles written by the authors. It is recommended to discuss the results with other studies directly related to the results of this article.
- As indicated above, include the information in table IV in the discussion.
- It would be useful to include in the discussion references to published studies on orthodontic force damage and to compare them with the results obtained in order to relate the clinical performance of the orthodontist to the possible damage that may be caused by his or her treatment.
Our response:
- We thank the reviewer for his/her concern and comments. We do hope that our changes are according to the reviewer‘s remarks.
- Revised text: entire Discussion section
The word did not keep the accuracy of the figure. We tried to insert once more the high-resolution picture, but the high resolution was not kept. We already send the high-resolution files.
- Revised text: Discussion section, pg. 15, lines 366-381
“Table 4 display the apical and cervical stress mostly in intact PDL when different failure criteria are employed. Here simulation employed all five criteria and compared the results with other studies [1, 2, 3, 12, 13, 15-17, 19-25, 28, 30] for investigating if significant differences are present. The reported quantitative values are comparable for studies using the same criterion (with small exceptions due to anatomical models and boundary conditions). However, the differences between the results of studies employing different failure criteria are significant. The closet is Von Mises and Tresca, which also provide results comparable with clinical data. The S1, S3 and hydrostatic pressure supply comparable quantitative reports, but higher than VM and Tresca, exceeding the 16 KPa of MHP (indirect validation criteria) even for no bone loss. Moreover, the hydrostatic pressure criteria studies (Wu et al. [15, 19, 20]) provided variable optimal amounts of force for intact PDL (i.e., 0.28-3.31 N) for different teeth (canine, premolar, lateral incisive), but with significant differences for the same tooth between similar reports (e.g., for canine: 1.7-2.1 N [15] and 3.31 N [19] for rotation and 0.38-0.4 N [15] and 2.3-2.6 N [20] for extrusion). For the premolar with no bone loss, the reported optimal rotational force was 2.8-2.9 N [19] opposingly to 0.5-0.6 N for intact PDL and 0.1-0.2 N 4-8 mm bone loss in present study.”
- Revised text: Discussion section, pg. 16, lines 412-418
“This approach agrees with clinical studies of orthodontic movements effects over the dental pulp [31-35] (i.e., forces of 0.15-1.5 N seemed to produce only circulatory disturbances and limited histological changes, with little evidence of severe ischemia and necrosis). Nonetheless, the same studies [31-35] reported periodontal injury to display a high risk of ischemia, necrosis and resorption, suggesting the need of using lower amounts of force, in agreement with herein observations.”
Concern of the reviewer: Conclusions
It is recommended that the number of conclusions be reduced by grouping conclusions related to the same topic under a single point.
Our response:
- We thank the reviewer for his/her concern and comments. We do hope that our changes are according to the reviewer‘s remarks.
- Revised text: entire Conclusions section, pg. 17, lines: 483-505
“The present findings show that:
- 0.5 N of orthodontic force is safe of any ischemic and necrotic risks up to 8 mm of periodontal breakdown, with rotation the most invasive (translation the least), and apical NVB more prone to ischemia than dental pulp. However, limited ischemic areas are also visible in coronal and radicular pulp for translational and rotational movements.
- A clear difference between failure criteria when analyzing the dental pulp and NVB cannot be proved based only on the correlation between the FEA quantitative results and MHP. Nonetheless, the difference could rely on type of analyzed material (i.e., ductile vs. brittle, with dental pulp, NVB and periodontal ligament having ductile resemblance), thus the choice should consider this issue. Tresca and VM failure criteria (adequate for the analysis of ductile materials) supplied comparable results and the lowest amount of stress.
- To obtain a clear image of the biomechanical behavior of tissues under orthodontic movements in reduced periodontium, correlations with the other components of periodontium, such as PDL should be considered.”
